# Synthetic RNA-based logic computation in mammalian cells

Satoshi Matsuura[1,2], Hiroki Ono[1,2], Shunsuke Kawasaki[1], Yi Kuang[1,3], Yoshihiko Fujita[1] & Hirohide Saito [1]

Synthetic biological circuits are designed to regulate gene expressions to control cell function. To date, these circuits often use DNA-delivery methods, which may lead to random genomic integration. To lower this risk, an all RNA system, in which the circuit and delivery method are constituted of RNA components, is preferred. However, the construction of complexed circuits using RNA-delivered devices in living cells has remained a challenge. Here we show synthetic mRNA-delivered circuits with RNA-binding proteins for logic computation in mammalian cells. We create a set of logic circuits (AND, OR, NAND, NOR, and XOR gates) using microRNA (miRNA)- and protein-responsive mRNAs as decision-making controllers that are used to express transgenes in response to intracellular inputs. Importantly, we demonstrate that an apoptosis-regulatory AND gate that senses two miRNAs can selectively eliminate target cells. Thus, our synthetic RNA circuits with logic operation could provide a powerful tool for future therapeutic applications.

[1] Department of Life Science Frontiers, Center for iPS Cell Research and Application (CiRA), Kyoto University, Kyoto 606-8507, Japan. [2] Graduate School of Medicine, Kyoto University, Kyoto 606-8501, Japan. [3]Present address: Department of Chemical and Biological Engineering, Hong Kong University of Science and Technology, Clear Water Bay, Hong Kong, SAR, China. Correspondence and requests for materials should be addressed to H.S. (email: hirohide.saito@cira.kyoto-u.ac.jp)

Synthetic biology approaches in mammalian cells have rapidly progressed in a variety of fields, suggesting great potential in medical applications including drug discovery, vaccine production, disease diagnosis, and cell therapy[1–4]. For example, researchers have designed several synthetic circuits that interface with endogenous gene networks to control apoptosis, differentiation, cell proliferation, and cell–cell communication[5–8]. For future therapeutic applications, it is important to improve the safety and specificity of the circuits, especially for the purpose of cell therapy in the field of regenerative medicine. A delivery method using modified messenger RNAs (modRNAs) could provide safer means to control gene expressions compared with DNA delivery, because modRNAs exhibit a short half-life in cells and do not cause random genomic integration[9–12].

To reduce off-target effects in non-target cells, it is important to produce desired outputs dependent on the cell state. One strategy is to design systems that determine the output of the circuits by sensing cell-specific, intracellular molecules as inputs. MicroRNAs (miRNAs) are a class of small noncoding RNAs that post-transcriptionally regulate gene expression by binding to target mRNAs[13,14]. It has been reported that more than 2600 different miRNAs exist in humans (miRBase ver.22)[15]. The miRNA expression profile is related to important biological processes, including development[16], cancer, and cell reprogramming[17,18], and thus can be used to classify the cell state[5,19–21]. These properties suggest that miRNA-responsive, synthetic circuits could provide useful tools for future therapeutic applications. We have previously designed miRNA-responsive, synthetic mRNAs (miRNA switches) that enable the detection and purification of target cells differentiated from human pluripotent stem cells based on endogenous miRNA activity[9,22]. However, the information from a single miRNA input may be insufficient to distinguish cells in a heterogeneous population. In these cases, it is crucial to detect multiple miRNA inputs and logically control the outputs (e.g. cell fate). Although synthetic circuits using modRNAs that encode RNA binding proteins (RBPs) have been constructed in mammalian cells[10], complex synthetic RNA-delivered circuits that can detect multiple miRNAs and regulate output protein through logic computation have not been demonstrated. Thus, we aimed to design synthetic RNA-delivered logic circuits that function in mammalian cells by improving the performance of miRNA- and protein-responsive modRNAs.

In this study, we construct a set of RNA-based logic circuits with RBPs that detect multiple miRNA inputs and regulate the output protein expression (Fig. 1a). We create five logic gates (AND, OR, NAND, NOR, and XOR) in mammalian cells using an RNA-only delivery approach. A 3-input AND circuit produces the output protein only in the presence of all target miRNAs. Additionally, we selectively control cell-death pathways between target and non-target cells by connecting a 2-input AND gate with apoptotic regulatory circuits.

## Results

**Improving the performance of miRNA-responsive circuits.** RBPs can function as both the input and the output of RNA-based regulatory devices[10]. For example, L7Ae, a kink-turn (Kt) RNA binding protein, associates with the Kt of archaeal box C/D sRNAs[23,24]. An L7Ae-Kt interaction at the 5′-UTR efficiently inhibits translation of the mRNA (Supplementary Figure 1b, d, f), probably by blocking translation initiation and ribosome function[25,26]. We have previously used the L7Ae-Kt interaction to construct modRNA-based regulatory devices that detect one target miRNA and regulate the production of one output protein[10]. The circuit topology of this device consists of two types of modRNAs (Fig. 1b); one is an *L7Ae*-coding mRNA with four

miRNA target sites that are completely complementary to the mature miRNA within the 3′ untranslated region (3′-UTR), and the other is an output-gene-coding mRNA with a Kt motif within the 5′-UTR. We refer to this device as L7-4xTX, where 4x represents the number of miRNA target sites, TX represents target sites to the specific miRNA, and the position of TX in the device name represents the location of the target site in the device (i.e., 5′-UTR or 3′-UTR). In the absence of the input miRNAs, the circuit produces no output protein due to L7Ae expression (OFF state), but produces the output protein in the presence of the input (ON state). However, the fold-change of the designed circuit between ON state and OFF state was moderate. As a first step toward realizing robust logic circuits with modRNAs, we aimed to improve the fold-change (ON/OFF ratio in output level) by enhancing sensitivity to the input miRNAs and reducing leaky protein expression, which would lead to a higher output expression level in ON state. We found that the knockdown effect of miRNAs on the miRNA switch is high when the target site (antisense sequence of the miRNA) was inserted into the 5′-UTR[9,22] (Supplementary Figure 1a, c, e). Thus, we hypothesized that the insertion of a miRNA target site into both the 5′-UTR and 3′-UTR may have a stronger effect than insertion in only one UTR and thus improve the fold-change between ON state and OFF state (Fig. 2a). Accordingly, we constructed *L7Ae*-coding modRNAs with miRNA target sites within both the 5′-UTR and 3′-UTR, and *EGFP*-coding reporter modRNA with a Kt motif within the 5′-UTR. We tested the performance of this miR-21-responsive RNA device by co-transfecting it with miR-21 mimic (chemically modified RNA that mimics endogenous miRNA) into 293FT cells. *iRFP670*-coding modRNAs without the miRNA target sites were also introduced as a transfection control. In this study, we chose miR-21 along with miR-302a as representative miRNA markers, because they are highly expressed in several human cancer cells[27] and pluripotent stem cells[28,29], respectively. We expected that the EGFP expression level would increase in a miR-21 mimic-dependent manner because 293FT cells do not express endogenous miR-21[9,30]. We used 8 nM miR-21 mimic because the proportional activity of 8 nM miR-21 mimic in 293FT cells (up to 15.3-fold, Supplementary Figure 1c) is almost equal to that of endogenous miR-21 in HeLa cells (up to 15.8-fold, Supplementary Figure 2), indicating that 8 nM miR-21 mimic reflects naturally occurring miRNA activity. Twenty-four hours after the transfection, we observed the circuit performance by flow cytometry analysis. We found that circuits with the device that contained miRNA target sites within both 5′-UTR and 3′-UTR (T21-L7-4xT21) showed the highest fold-change (9.2-fold) compared with standard circuits containing miRNA target sites only within the 3′-UTR (L7-4xT21, 2.7-fold) or within the 5′-UTR (T21-L7, 5.2-fold) (Fig. 2b–d). The circuit with T21-L7-4xT21 modRNA was much more effective at distinguishing cell populations in ON and OFF states compared with the other modRNA devices (Fig. 2c).

To investigate whether the T21-L7-4xT21 circuit can detect endogenous miRNAs, we transfected it into HeLa cells, which express endogenous miR-21[9,30]. The increased fold-change and cell separation between ON and OFF (with 8 nM miR-21 inhibitor) states with the T21-L7-4xT21 circuit was confirmed in HeLa cells (from 1.1- to 3.1-fold) (Fig. 2e, f). In addition, we used the T302a-L7-4xT302a circuit to detect another type of miRNA (miR-302a). We confirmed a significant fold-change between ON and OFF states (from 4.6 to 9.0-fold) in 293FT cells (Supplementary Figure 3). The results were consistent with those for miR-21 (Fig. 2), confirming that the improvement of the circuit performance by using modRNAs that contain the miRNA target site within both 5′- and 3′-UTR is independent of the miRNA sequence or cell line.

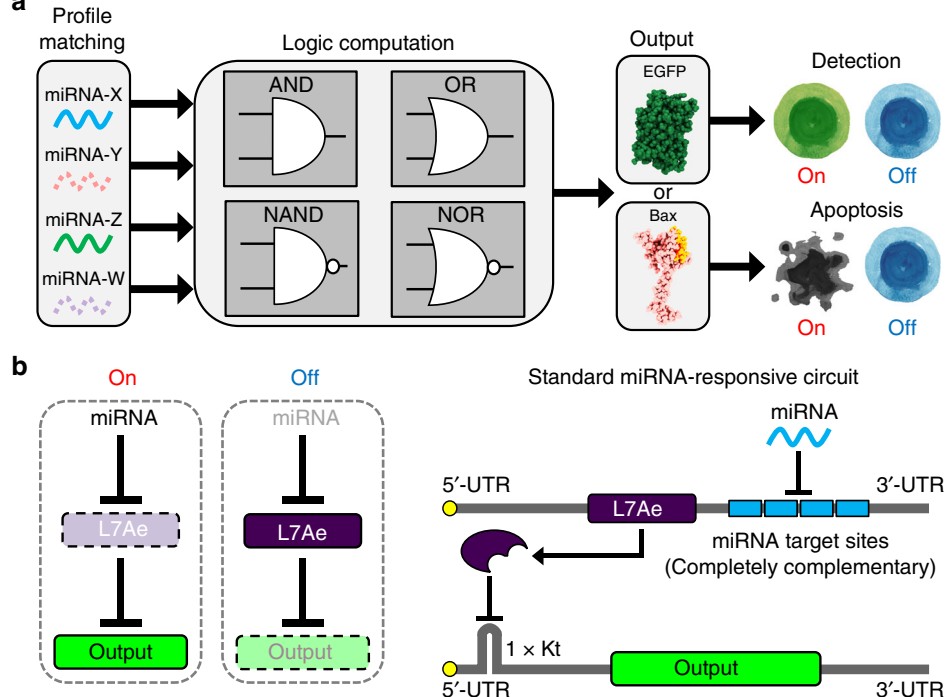

**Fig. 1** Schematic representation of the miRNA-responsive circuit and its topology. **a** Logic circuits can be constructed to detect cell-specific multiple miRNAs as input and translate genes (*EGFP* or *hBax* in this study) as output only in target cells. **b** In OFF state (absence of input miRNAs), L7Ae protein represses translation of the output gene-coding mRNAs by interacting with the kink-turn motif (Kt). In ON state (presence of input miRNAs), the L7Ae translation is repressed by the miRNAs, which leads to output translation

**Construction of logic circuits using modRNA-delivered device**. Next, we constructed a modRNA-deliverable set of logic circuits that can sense the activities of multiple miRNAs to regulate output protein production. First, we designed a 2-input (miR-21 and miR-302a) AND circuit with EGFP as the output (first row in Fig. 3). The AND circuit expresses the output only in the presence of both miRNAs (input pattern [11] in the truth table of Fig. 3). The circuit consists of miR-21- or miR-302a-responsive *L7Ae*-coding mRNAs, and *EGFP*-coding mRNA with a Kt motif (Kt-EGFP). We tested the performance of the circuit in 293FT cells, which have no activity for either miRNA, thereby enabling examination of the conditional four input patterns (denoted as [00], [10], [01], and [11]) by treatment with miR-21 (302a) mimics. As expected, the AND circuit functioned only in the presence of both miRNAs (Fig. 3 and Supplementary Figure 4). Because L7Ae efficiently represses translation of Kt-EGFP mRNA[25,26] (Supplementary Figure 1b, d, f), we found that the presence of either *L7Ae*-coding mRNA was sufficient to repress EGFP expression. The fold-change was calculated by dividing the averaged output levels in each ON state ([11] for AND gates) by that in each OFF state ([00], [10] and [01] for AND gates) and found it to be 7.1-fold (Fig. 3). We next designed an OR circuit (second row in Fig. 3), which expresses the output when either one or both inputs are present (ON state = [10], [01], or [11]). This circuit consists of miR-21- and/or miR-302a-responsive single mRNA and *EGFP*-coding mRNA with a Kt motif. The designed OR circuit functioned with a fold-change of 5.4 (Fig. 3).

To generate more complex circuits, we next used a bacteriophage MS2 coat protein, MS2CP[31], as a second translational repressor protein in addition to L7Ae. First, we improved the response of MS2CP-responsive mRNA by engineering the surrounding sequence containing the binding motif (MS2box). The sc2xMS2box motif-inserted mRNA, which consists of two MS2box motifs and a scaffold structure to stabilize the MS2CP-

binding motif[32], showed the highest fold-change (14.7-fold) compared with that of other MS2CP-responsive mRNAs, which had one (1.85-fold, 1xMS2box) or two (2.8-fold, 2xMS2box) MS2box motifs inserted into the 5′-UTR (Supplementary Figure 5). From this inclusion, we designed NAND, NOR, and XOR circuits (Fig. 3) by connecting L7Ae- and MS2CP-responsive mRNAs. NAND (Not AND) and NOR (Not OR) circuits can be designed by inverting the output of AND and OR circuits, respectively. We used *MS2CP*-coding mRNAs with a Kt motif (Kt-MS2CP) and *EGFP*-coding mRNA with two MS2CP binding motifs within the 5′-UTR (sc2xMS2box-EGFP) as a second repressor device. The NAND circuit should produce no output only if both input miRNAs are present ([11]). The NOR circuit should produce outputs only when both inputs are absent ([00]). The XOR (eXclusive OR) circuit produces outputs only when exactly one input miRNA is present ([10] or [01]). Our NAND, NOR, and XOR circuits worked as expected, with fold-changes of 3.1, 3.5, and 5.5, respectively (Fig. 3). From these results, we confirmed all the designed basic circuits (AND, OR, NAND, NOR, XOR) worked in mammalian cells using a modRNA-delivery approach (Fig. 3 and Supplementary Figure 4).

In addition, we designed a 3-input AND circuit using miR-21-, miR-302a-, and miR-206-responsive, *L7Ae*-coding mRNAs with Kt-EGFP mRNAs (Fig. 4a). As expected, the circuit produced output EGFP production only in the presence of all three miRNAs, with a fold-change of 4.4 (Fig. 4b and Supplementary Figure 6).

**Apoptosis regulatory 2-input AND circuit**. Finally, we validated whether RNA-based circuits can control cell-death signals through a logic operation. We designed a 2-input (miR-206 and miR-302a) AND circuit with human *Bax* (hBax), a pro-apoptotic gene, as the endpoint output. In addition, *Bcl-2*, an anti-apoptotic

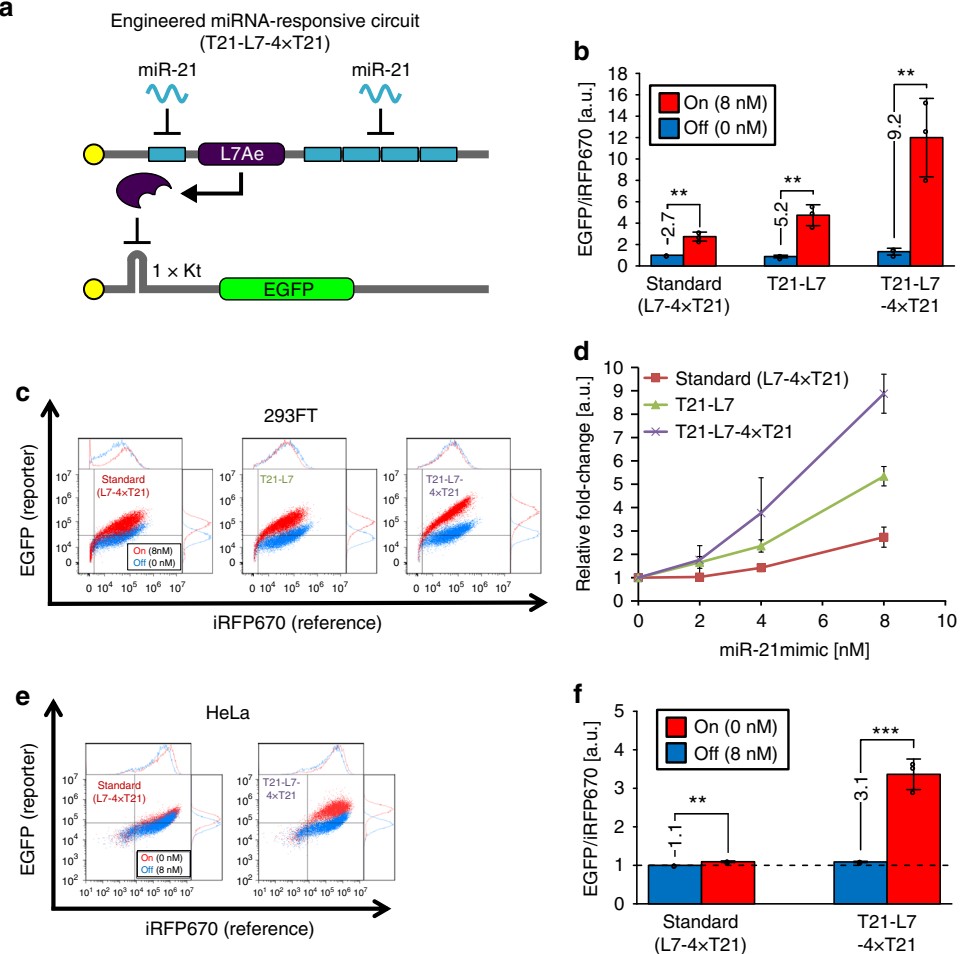

**Fig. 2** Engineering of L7Ae-mediated miR-21 circuits. **a** Schematic representation of engineered L7Ae-mediated, miR-21-responsive circuits with miRNA target sites within the 5'- and 3'-UTR. **b**, **f** The bar charts represent the gene translation output (EGFP/iRFP670) for each circuit in 293FT cells (**b**) and in HeLa cells (**f**). *Y*-axis represents the mean EGFP per mean iRFP670 signal (EGFP/iRFP670). The output levels in OFF states of the standard circuit were normalized to 1. ON states in 293FT cells show the cells co-transfected with each circuit and miR-21 mimic (8 nM) (**b**). miR-21 inhibitor (8 nM) was used to generate OFF states in HeLa cells (**f**). Data are represented as the mean ± s.d. (*n* = 3). **c**, **e** Representative scatter plots and histograms for each circuit in 293FT cells (miR-21 mimic 0 or 8 nM) (**c**) and in HeLa cells (miR-21 inhibitor 0 nM or 8 nM) (**e**). Gates shown on the plots were generated by negative control (non-transfected) cells. ON state and OFF state (red and blue) show the results in the presence and absence of input miRNAs, respectively. **d** Dose–response curves of the engineered L7Ae-mediated miRNA circuits with miR-21 mimics ranging from 0 to 8 nM in 293FT cells. The EGFP/iRFP670 ratio in each circuit without miR-21 mimic was set to 1. Data are shown as the mean ± s.d. (*n* = 3). The levels of significance (unpaired two-tailed Student's *t*-test) are denoted as **P < 0.01 and ***P < 0.001. a.u. arbitrary units, iRFP near-infrared red fluorescent protein. Transfection details are described in Supplementary Table 4

gene was fused with *L7Ae* through P2A peptides to reinforce the repression of apoptosis against leaky hBax expression in OFF states (Fig. 5a, b). In this design, we expected that the circuits should kill cells only in the presence of both target miRNAs ([11] state). We co-transfected the circuits with miR-206 and/or miR-302a mimics into 293FT cells. Twenty-four hours after the transfection, we stained the cells with SYTOX red for dead cells and Annexin V for apoptotic cells to quantitatively assess the apoptosis level. The circuits induced apoptosis only when both input miRNAs were present. The apoptosis level in ON state was comparable to *hBax* mRNA transfection (Fig. 5c, d). Thus, our apoptosis regulatory 2-input AND circuit can selectively regulate cell death by sensing two target miRNAs.

## Discussion

In this study, we designed and constructed multi-input logic circuits that can distinguish differences in the activities of

multiple miRNA in a cell with an RNA-only delivery approach. We found that several basic logic circuits (AND, OR, NAND, NOR and XOR gates) can be constructed with a set of RBPs and mRNAs without the requirement of DNA-based transcriptional regulation by improving the performance of RNA regulatory devices (Figs. 3 and 4). To quantitatively evaluate the performance of each circuit, we calculated the cosine similarity and net fold-change (Supplementary Figure 7). Cosine similarity is an index for evaluating the error between the ideal implementation and observed behavior in a circuit[33] (see Analysis of cosine similarity and net fold-change in Methods). The net fold-change was defined as the ratio of the averaged output level in ON and OFF states. From the analyses, AND and OR circuits showed better performance in net fold-change and cosine similarity, respectively, compared with NAND and NOR circuits (Supplementary Figure 7), which we attribute to circuit complexity. All of the circuits showed statistically significant performance (Supplementary Tables 1, 2, and 3).

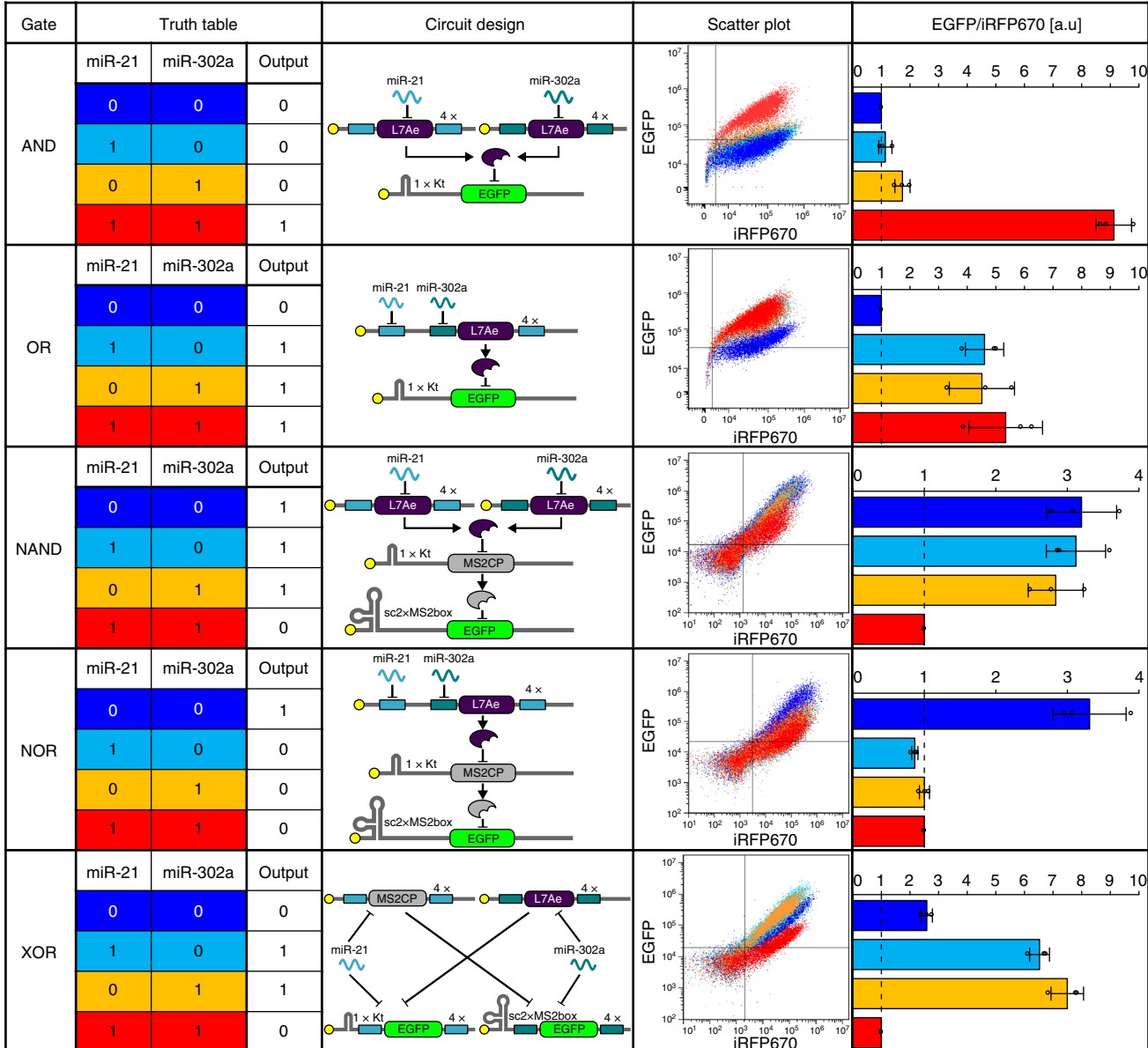

**Fig. 3** 2-input miRNA-responsive logic circuits. Different colors correspond to those in the truth table, scatter plot and output level columns, respectively. MiR-21 and miR-302a mimics were used as inputs. For example, input pattern [10] colored in light blue means miR-21 present (8 nM) and miR-302a absent (0 nM). The scatter plot column shows the overlay of scatter plots for four input patterns in each circuit. The output level was calculated by dividing the mean EGFP signal by the mean iRFP670 signal. The ratio (EGFP/iRFP670) was normalized by the ratio corresponding to a representative OFF state for each circuit. Data are shown as the mean ± s.d. ($n = 3$). Tukey's multiple comparisons test results are shown in Supplementary Table 1

Although DNA-based circuits have great potential for applications such as designer cells[7,34] (e.g., CAR-T cells), RNA-delivered circuits have an advantage in terms of safety, which makes them preferable for therapeutic applications in the field of regenerative medicine, such as the elimination of unwanted cells and purification of target cells from a heterogenous population differentiated from human pluripotent stem cells[9,22]. MicroRNA-responsive circuits will be especially useful, because miRNA expressions are signatures of the cell identity and cell state. However, to date, most studies using synthetic circuits have required a DNA delivery method[5,35–38]. We and others have developed synthetic RNA-delivered, miRNA-responsive circuits[10]. To control cell fate more precisely, multi-input circuits with logic computation is necessary, as demonstrated in DNA-based systems[5,33,39,40]. However, the construction of logic circuits in cells with RNA-only delivery has not been achieved previously,

because previous RNA-based circuits show relatively low fold-change (ON/OFF ratio in outputs) and have a limited repertoire of devices with high repression capacity. By improving sensitivity to the input miRNA (Fig. 2) and engineering a repressor device (MS2CP-responsive mRNA) to increase the ON/OFF ratio (Supplementary Figure 5), here we report five kinds of 2-input basic logic circuits compatible with RNA-only delivery. These basic logic circuits which are composed of simple two types of repressor (miRNA- and protein-responsive mRNAs) are an important milestone toward the construction of scalable and more complex RNA-only circuits.

To further engineer and improve synthetic RNA-based circuits that can respond to multiple miRNA targets, three issues regarding the circuit design should be considered. First, to increase the fold-change between ON and OFF states, we need to reduce undesired leakiness in the protein expression prior to

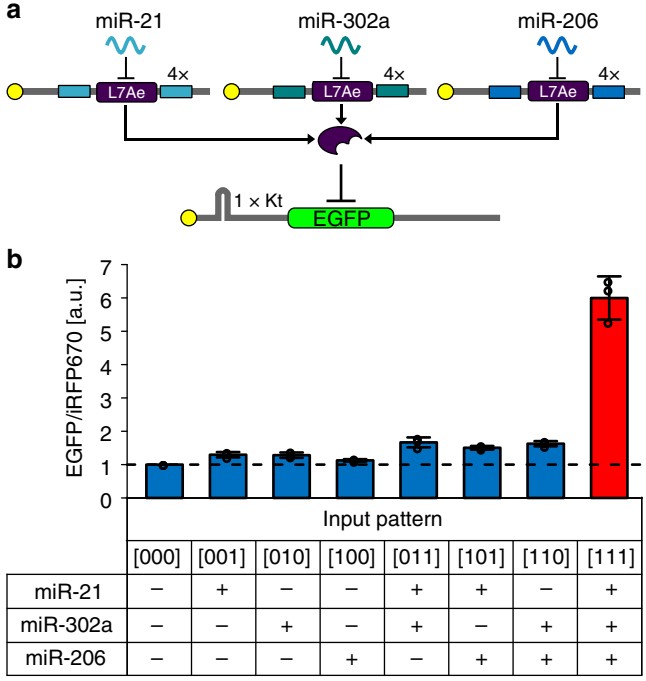

**Fig. 4** 3-input AND circuit. **a** The circuit recognizes the activity of three types of miRNAs (miR-21, miR-302a, and miR-206) and produces high output (EGFP) only when all three miRNAs are present ([[111] state). The circuit design is shown in **a**. **b** The bar chart represents the output level (EGFP/iRFP670) for each state in 293FT cells. Data are shown as the mean ± s.d. (n = 3). Tukey's multiple comparisons test results are shown in Supplementary Table 2

miRNA-mediated post-transcriptional repression. For this purpose, the circuit may benefit by adding a post-translational system that controls the stability of the output protein products (such as the degron system[41]). As an alternative approach, the insertion of multiple miRNA target sites into the 5′-UTR of the sensor mRNAs may enhance the miRNA sensitivity and circuit performance[30]. Although we do not know the reason why the insertion of a miRNA target site into the 5′-UTR and 3′-UTR together results in enhanced downregulation of the target mRNAs[42], we assume that each designed miRNA target site is completely complementary to the miRNA of interest, which would thus induce AGO2-mediated mRNA cleavage[43]. Second, to perform more complexed logic computation in cells, we need to scale-up the RNA-based circuits by using a set of orthogonal RBPs. Currently, the limited availability of translational repressors, such as L7Ae or MS2CP, makes it difficult to design more complex circuits. Finding new RBP-mediated repressors, such as CRISPR-Cas effector Cas13, may expand the repertoire of RNA-based circuits[44–46]. Lastly, it is important to choose appropriate input miRNAs in order to detect and control the target cells in a heterogenous population. For this purpose, we need to measure not only the expression profiles, but the activity profiles of the miRNAs, because many miRNAs detected by RNA sequencing or microarray have weak or little activity according to reporter analyses[42,47,48].

In conclusion, we have developed a framework for constructing basic logic circuits with RNA-only delivery, expanding the potential of RNA-based gene circuits to detect and control the cell state. We demonstrated that 2-input AND circuit with an apoptotic gene as the output regulated cell death according to differences in the two miRNA activities. Such a multi-input system enables us to purify target cells or control cell fate more

precisely compared with a single miRNA-input system[9,22]. Synthetic mRNA-based, multi-input miRNA-responsive circuits will contribute to the development of more sophisticated circuits for future medical applications.

## Methods

**Cell culture**. 293FT cells (Invitrogen, USA) were cultured in DMEM high glucose (Nacalai Tesque, Japan) supplemented with 10% FBS (JBS, Japan), 0.1 mM MEM non-essential amino acids (Life Technologies, USA), 2 mM L-glutamine (Life Technologies) and 1 mM sodium pyruvate (Nacalai Tesque). HeLa CCL2 cells (ATCC) were cultured in DMEM High Glucose (Nacalai Tesque) supplemented with 10% FBS (JBS). All cell lines were cultured at 37 °C with 5% $CO_2$.

**RNA transfection**. All transfections were performed in 24-well format using Stemfect RNA Transfection Reagent Kit (Stemgent, USA) or Lipofectamine® MessengerMAX (Thermo Fisher scientific, USA) according to the manufacturer's protocol. Opti-MEM (Thermo Fisher scientific) was used as buffer for MessengaerMAX. The MessengerMAX reagent and buffer were mixed for 10 min. The mRNAs with or without miRNA mimics or miRNA inhibitors diluted with buffer were mixed with the above reagent for 5 min. 293FT cells (1 × 10⁵ cells per well) and HeLa cells (5 × 10⁴ cells per well) were seeded in 24-well plates at 24 h before the transfection for all experiments. The medium was not changed before and after the transfection. All subsequent assays were performed 24 h after the transfection. The transfection details for each experiment are shown in Supplementary Table 4.

**Preparation of DNA template for in vitro transcription (IVT)**. A DNA template for IVT was generated by PCR using KOD-Plus-Neo (TOYOBO, Japan). A forward primer containing the T7 promoter and a reverse primer containing 120-nucleotide-long poly(T) tract transcribed into a poly(A) tail were used. PCR products amplified from the plasmids were subjected to digestion by DpnI restriction enzyme (TOYOBO). The PCR products were purified using a MinElute PCR purification Kit (QIAGEN, UK) according to the manufacturer's protocol.

**Preparation of modified mRNA**. All mRNAs were generated using the above PCR products and MEGAscript T7 Kit (Ambion, USA). In the reaction, pseudouridine-5′-triphosphate and 5-methylcytidine-5′-triphosphate (TriLink BioTechnologies, USA) were used instead of uridine triphosphate and cytosine triphosphate, respectively. For IVT of the MS2CP-responsive mRNA in Fig. 3, N1-methylpseudouridine-5′-triphosphate (m1pU) (TriLink BioTechnologies) was used instead of uridine-5′-triphosphate. Guanosine-5′-triphosphate was 5-fold diluted with an anti-reverse cap analog (TriLink BioTechnologies) before the IVT reaction. Reaction mixtures were incubated at 37 °C for up to 6 h and then mixed with TURBO DNase (Ambion), and further incubated at 37 °C for 30 min to remove template DNA. The resulting mRNAs were purified using a FavorPrep Blood/Cultured Cells total RNA extraction column (Favorgen Biotech, Taiwan), incubated with Antarctic Phosphatase (New England Biolabs) at 37 °C for 30 min, and then purified again using an RNeasy MinElute Cleanup Kit (QIAGEN).

**Synthetic miRNA mimics and inhibitors**. MiRNA mimics are small, chemically modified double-stranded RNAs that mimic endogenous miRNAs. The RNA mimic of human miR-21-5p, miR-302a-5p and miR-206 and negative control miRNA were used (Thermo Fisher Scientific). The negative control mimic is a random sequence validated not to have any downstream mRNA target for repression. MiRNA inhibitors for miR-21-5p (Thermo Fisher Scientific) were used in experiments using HeLa cells.

**Fluorescent microscopy**. Fluorescent images were taken at 24 h after the transfection by IX81 microscopy connected to a CDD-camera (Olympus, Japan). Images were edited to change the brightness and contrast using ImageJ software (NIH, Bethesda, MD, USA).

**Flow cytometry and data analysis**. All flow cytometry measurements were performed 24 h after the transfection using BD Accuri™ C6 (BD Biosciences, USA). For all fluorescence assays, clumps and doublets were excluded based on forward and side scatter. EGFP and iRFP670 were detected by FL1 (533/30 nm) and FL4 (675/25 nm) filters, respectively. The data were analyzed using FlowJo 7.6.5 software.
The output level was calculated by the following formula:

$$\frac{\text{Mean EGFP intensity in iRFP670}^+ \text{ cells}}{\text{Mean iRFP670 intensity in iRFP670}^+ \text{ cells}} \quad (1)$$

iRFP670⁺ gating was determined from the mock sample with 99.9% cells outside the gate. Each data set was normalized by an appropriate control sample and then averaged by 3 data sets.

**Apoptosis and cell death assays**. Sample cells including those in the supernatant were collected 24 h after the transfection, washed with PBS and stained with

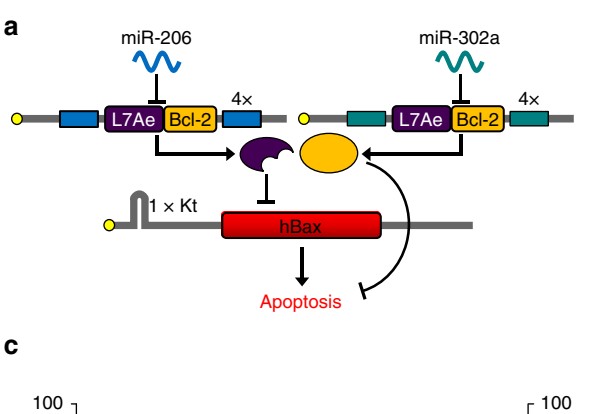

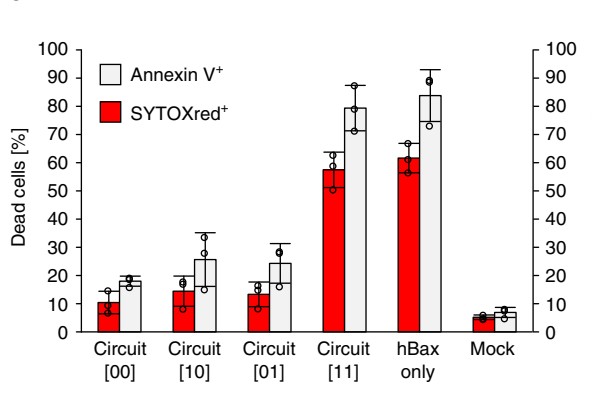

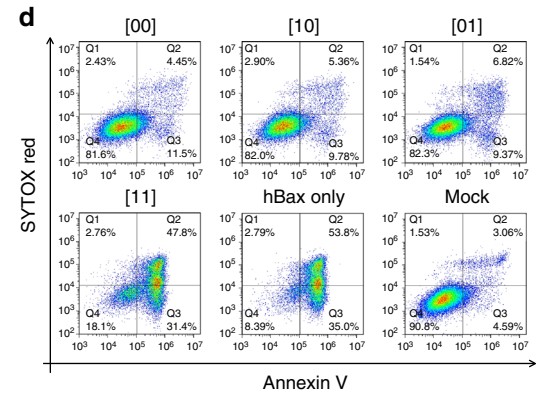

% of dead cells = Q1+Q2

% of apoptotic cells = Q2+Q3

**Fig. 5** Apoptosis regulatory 2-input AND circuit. **a** The circuit has a pro-apoptotic gene, *hBax*, as the output. The anti-apoptotic gene *Bcl-2* was fused with the *L7Ae* gene through P2A peptides to enhance the repression of apoptosis. **b** The truth table in the circuit is shown. For example, input pattern [10] means miR-206 present (=8 nM) and miR-302a absent (=0 nM). The circuit induces apoptosis (cell death) as output only when both miRNAs are present (=[11] state). **c** Cells were stained with SYTOX red for dead cell staining and Annexin V for apoptotic cell staining 24 h after the transfection. Data are represented as the mean ± s.d. (n = 3). Tukey's multiple comparisons test results are shown in Supplementary Table 3. **d** Representative flow cytometry data of apoptosis regulatory AND circuits. The positive rate of SYTOX red was calculated by the sum of the percentage of the Q1 and Q2 fraction. The positive rate of Annexin V was calculated by the sum of the percentage of the Q2 and Q3 fraction

Annexin V, Alexa Fluor 488 conjugated (Life Technologies) and SYTOX red (Life Technologies) for 15 min at room temperature. The cells were analyzed by BD Accuri™ C6. Annexin V, Alexa Fluor 488 conjugate was detected with FL1 filter, and SYTOX red dead-cell staining was detected with FL4 filter.

**Analysis of cosine similarity and net fold-change**. In Supplementary Figure 7, the correctness of multi-input logic circuits was quantitatively evaluated by calculating the cosine similarity between vectors **x** and **y** using the following formula:

$$\cos\theta = \frac{\mathbf{x}\cdot\mathbf{y}}{|\mathbf{x}||\mathbf{y}|} \tag{2}$$

**x** is a truth table vector that has ideal output (=0 or 1) for each state ([00], [10], [01], and [11]) as a vector. For example, **x** = (0 0 0 1) for AND circuit. **y** is an output signal vector that carries the observed output levels (=EGFP/ iRFP670) of each state ([00], [10], [01], and [11]). Thus, cos θ ranges from 0 (worst) to 1 (best). Net fold-change was calculated by dividing the averaged output level in each ON state by that in each OFF state.

**Statistical analysis**. All data are presented as the mean ± s.d. Unpaired two-tailed Student's *t*-test was used for the statistical analysis in Fig. 2 and Supplementary Figure 3. Tukey's method was used for the statistical analysis in Figs. 3–5 (Supplementary Tables 1, 2 and 3). The levels of significance are denoted as *P < 0.05, **P < 0.01, ***P < 0.001, ****P < 0.0001, and n.s., not significant (P ≥ 0.05). All statistical tests were performed using R.

## Data availability
All relevant data are available from the corresponding author upon reasonable request. Primer sequences are provided in Supplementary Table 5.

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

## Acknowledgements

We thank Saito laboratory members for kind advice about the experimental conditions, data analysis, and discussion. We also thank Dr. Peter Karagiannis (Kyoto University) and Ms. Yukiko Nakagawa and Miho Nishimura for critical reading of the manuscript and administrative support, respectively.

## Author contributions

S.M., Y.F., and H.S. conceived the project and designed the experiments. S.M. performed all the experiments except for Supplementary Figure 5. S.K. and Y.K. designed the MS2CP-responsive mRNAs. H.O. supported the experiments in Fig. 2e, f and Supplementary Figure 5. S.M., Y.F., and H.S. wrote the manuscript. All authors discussed the results and commented on the manuscript.

## Additional information

**Competing interests:** The authors declare no competing interests.

