## [Peer Review File · Nature Communications]

Reviewers' Comments:

Reviewer #1:

Remarks to the Author:

The manuscript "Synthetic RNA-based logic computation in mammalian cells" introduces an RNA/RBP-based system of logic circuits designed to be triggered by multiple miRNA inputs in living cells. The authors demonstrate two-input AND, OR, NAND, NOR, and XOR gates in addition to three-input AND gates with outputs quantified by EGFP fluorescence and apoptosis.

Overall, this work introduces an interesting approach to synthetic RNA-based circuitry which has been previously limited to single input systems. The designs of the logic gates are original and achieve programmed outputs which are well-supported by statistical significance. As the latest development in RNA-based computation, this work has great implications for the future utilization of circuits for both diagnostic and therapeutic applications.

Major points:

1. The nomenclature used in this work should be clearly explained upfront.
2. Please explain the rationale of choosing miR-21 and miR-302a.
3. Did the authors consider designing their switches to mimic the siRNA action (fully complementary)? This could improve the response and reduce undesired leakiness in the protein expression, since the lower count of siRNAs would require for the complete deactivation.
4. In Figure 2: Why 8 nM was chosen as the highest concentration of miR-21? Were other concentrations (10, 20, etc) tested?
5. Why are there differences in outputs between Figure 2B (at 8 nM) and 2D? Would not you expect to see comparable ratios?
6. For experiments with HeLa cells (Figure 2f), was inhibitor used at 8 nM concentration? If so, how was this concentration chosen?
7. Transfection experiments are essential for this work, therefore more details for incubation times, buffers, media, possible media change, etc should be provided.
8. Statistical analysis should be performed for AND and XOR gates in Figure 3 and for the results in Figure 5b.

Minor point:

1. Though it is explained in the caption, Figure 5 could be clarified with a truth table.

Reviewer #2:

Remarks to the Author:

In this manuscript Matsuura and colleagues describe the further refinement of miRNA sensing RNA circuits. The work is well performed and described but I found that it did not make a significant advance over what has been previously published. The manuscript contains a small set of experiments that use miRNA target mRNAs containing coding sequences for the L7Ae RNA-binding protein, which in turn binds another mRNA encoding EGFP or Bax. The uses of L7Ae in applications in RNA synthetic biology have been described in a number of studies from this team already (e.g. Wroblewska et al. *Nat Biotechnol.* 2015; Stapleton et al. *ACS Synth Biol.* 2012; Saito et al. *Nat Commun.* 2011; and Saito et al. *Nat Chem Biol.*) and I cannot see that this offers much beyond what they and others (e.g. Quarton et al. *NPJ Syst Biol Appl.* 2018; Schreiber et al. *Mol Syst Biol.* 2016; Ehrhardt et al. *Biosens Bioelectron.* 2015; Lapique et al. *Nat Chem Biol.* 2014; Strovas et al. *ACS Synth Biol.* 2014; Haynes et al. *ACS Synth Biol.* 2012; and Xie et al. *Science.* 2011) have published in this area already. In particular the work is remarkably similar to that described in the 2015 *Nature Biotechnology* paper. Therefore, this study seems better suited to a more specialized journal.

Point By Point Response:

Reviewer #1

Reviewer #1 (Remarks to the Author):

The manuscript “Synthetic RNA-based logic computation in mammalian cells” introduces an RNA/RBP-based system of logic circuits designed to be triggered by multiple miRNA inputs in living cells. The authors demonstrate two-input AND, OR, NAND, NOR, and XOR gates in addition to three-input AND gates with outputs quantified by EGFP fluorescence and apoptosis.

Overall, this work introduces an interesting approach to synthetic RNA-based circuitry which has been previously limited to single input systems. The designs of the logic gates are original and achieve programmed outputs which are well-supported by statistical significance. As the latest development in RNA-based computation, this work has great implications for the future utilization of circuits for both diagnostic and therapeutic applications.

Response:

We thank the reviewer for the positive comments and helpful suggestions on how to improve the manuscript. We added new data and revised the text based on the suggestions as bellow.

Major points:

1. The nomenclature used in this work should be clearly explained upfront.

Response 1:

We apologize for confusion about the nomenclature. We added additional explanations about the nomenclature in the revised text and are consistent with the names of the constructs used as follows:

Page 4, line 125:

“We refer to this device as L7-4xTX, where 4x represents the number of miRNA target sites, TX represents target sites to the specific miRNA, and the position of TX in the device name represents the location of the target site in the device (i. e., 5'-UTR or 3'-UTR).“

Page 7, line 207:

“We used MS2CP-coding mRNAs with a Kt motif (Kt-MS2CP) and EGFP-coding mRNA with two MS2CP binding motifs within the 5'-UTR (sc2xMS2_{box}-EGFP) as a second repressor device. “

2. Please explain the rationale of choosing miR-21 and miR-302a.

Response 2:

Thank you for the comment. It is known that miR-21 and miR-302a are highly expressed in several cancer cells and human pluripotent stem cells (e.g., ESCs, iPSCs), respectively. Therefore, these miRNAs are promising targets for detecting or eliminating cancer-like cells or pluripotent stem cells for future applications using our circuits. In addition, these miRNAs are not expressed in 293FT cells. Therefore, using the miRNA mimics, we can manipulate the activity of these miRNAs in cells to validate the performance of the circuits. To explain the above points more clearly, we added an explanation into the revised Results as follows:

Page 5, line 146:

“In this study, we chose miR-21 along with miR-302a as representative miRNA markers, because they are highly expressed in several human cancer cells²⁷ and pluripotent stem cells^{28,29}, respectively.”

3. Did the authors consider designing their switches to mimic the siRNA action (fully complementary)? This could improve the response and reduce undesired leakiness in the protein expression, since the lower count of siRNAs would require for the complete deactivation.

Response 3:

Yes, we expected our system mimics siRNA action. We designed miRNA target sites with a completely complementary sequence for miRNAs to reinforce the response of the circuits. To emphasize this point, we added the following text and revised Figure 1b in the revised manuscript. To explain the above points, we added the new sentences in Results as follows:

Page 4, line 123:

“one is an L7Ae-coding mRNA with four miRNA target sites that are completely complementary to the mature miRNA and within the 3' untranslated region (3'-UTR), “

4. In Figure 2: Why 8 nM was chosen as the highest concentration of miR-21? Were other concentrations (10, 20, etc) tested?

Response 4:

We used 8 nM miR-21 mimic because the activity of 8 nM miR-21 mimic in 293FT cells is almost equal to that of endogenous miR-21 in HeLa cells, indicating that 8 nM is close to physiological condition. To confirm this point, we added new experimental data (compare new Supplementary

Figure 2a with Supplementary Figure 1c). Experiments using higher mimic concentration (more than 8 nM) may have shown a higher fold-change, however, as suggested above, these findings are unlikely physiologically relevant. One goal of this manuscript is to emphasize miRNA mimics for practical use. To emphasize this point, we have added the following text to the revised manuscript. To explain the above points, we added the new sentences in Results as follows:

Page 5, line 150:

“We used 8 nM miR-21 mimic because the proportional activity of 8 nM miR-21 mimic in 293FT cells (up to 15.3-fold, Supplementary Fig. 1c) is almost equal to that of endogenous miR-21 in HeLa cells (up to 15.8-fold, Supplementary Fig. 2), indicating that 8 nM miR-21 mimic reflects naturally occurring miRNA activity. “

5. Why are there differences in outputs between Figure 2B (at 8 nM) and 2D? Would not you expect to see comparable ratios?

Response 5:

We apologize for the confusion. The data in Figure 2b and 2d (at 8 nM) are the same data set but shown with different normalizations. In Figure 2b we normalized all six data by normalizing output levels with the OFF state of standard circuits (L7-4 × T21). In Figure 2d, the output levels in each circuit were normalized with 0 nM. That is, the Y-axis in Figure 2d shows the relative fold-change between three kinds of circuits, so the 9.2-fold shown in Figure 2b is equal to the value of T21-L7-4xT21a at 8 nM shown in Figure 2d. In the revised manuscript, we revised the legends of Figure 2 and Supplementary Figure 3.

6. For experiments with HeLa cells (Figure 2f), was inhibitor used at 8 nM concentration? If so, how was this concentration chosen?

Response 6:

Yes, the inhibitor concentration in Figure 2f was 8 nM. We chose 8 nM, because this concentration is sufficient for repressing endogenous miR-21 activity in HeLa cells. To confirm this point, we added new data as Supplementary Figure 2 on the co-transfection of miR-21 switch and miR-21 inhibitor with different concentrations into HeLa cells. As shown in Supplementary Figure 2a, the EGFP/iRFP670 level of miR-21 switch at 8 nM inhibitor equals that of control mRNA switch.

7. Transfection experiments are essential for this work, therefore more details for incubation times, buffers, media, possible media change, etc should be provided.

Response 7:

As suggested, we added the detailed transfection procedure in the subsection RNA transfection of Material and Methods.

Page 10, line 320:

“Opti-MEM (Thermo Fisher scientific) was used as buffer for MessengerMAX. The MessengerMAX reagent and buffer were mixed for 10 minutes. The mRNAs or miRNA mimics or miRNA inhibitor diluted with buffer were mixed with the above reagent for 5 minutes. 293FT cells (1×10^5 cells/well) and HeLa cells (5×10^4 cells/well) were seeded in 24-well plates at 24 h before the transfection for all experiments. The medium was not changed before and after the transfection. All subsequent assays were performed 24 h after the transfection.”

8. Statistical analysis should be performed for AND and XOR gates in Figure 3 and for the results in Figure 5b.

Response 8:

We performed statistical analysis for all the logic gates in Figure 3 and show the data in Supplementary Table 1 of the revised manuscript. As the reviewer pointed out, we did not perform statistical analysis for the logic circuits shown in Figure 5b. In the revised manuscript, we performed statistical analysis for the data in new Fig. 5c (previous Fig. 5b) and Figure 4. The analysis showed that the performance of both circuits is statistically significant. We added these statistical data in Supplementary Tables 2 and Table 3.

Minor point:

1. Though it is explained in the caption, Figure 5 could be clarified with a truth table.

Response:

Thank you for the suggestion. As suggested, we made a truth table and added it in Figure 5.

Reviewer #2

Reviewer #2 (Remarks to the Author):

In this manuscript Matsuura and colleagues describe the further refinement of miRNA sensing RNA circuits. The work is well performed and described but I found that it did not make a significant advance over what has been previously published. The manuscript contains a small set of experiments that use miRNA target mRNAs containing coding sequences for the L7Ae RNA-binding

protein, which in turn binds another mRNA encoding EGFP or Bax. The uses of L7Ae in applications in RNA synthetic biology have been described in a number of studies from this team already (e.g. Wroblewska et al. *Nat Biotechnol.* 2015; Stapleton et al. *ACS Synth Biol.* 2012; Saito et al. *Nat Commun.* 2011; and Saito et al. *Nat Chem Biol.*) and I cannot see that this offers much beyond what they and others (e.g. Quarton et al. *NPJ Syst Biol Appl.* 2018; Schreiber et al. *Mol Syst Biol.* 2016; Ehrhardt et al. *Biosens Bioelectron.* 2015; Lapique et al. *Nat Chem Biol.* 2014; Strovas et al. *ACS Synth Biol.* 2014; Haynes et al. *ACS Synth Biol.* 2012; and Xie et al. *Science.* 2011) have published in this area already. In particular the work is remarkably similar to that described in the 2015 *Nature Biotechnology* paper. Therefore, this study seems better suited to a more specialized journal.

Response:

We apologize for our insufficient explanation about the advance over previous works. There are two points we would like to consider. First is the advantages of RNA-based logic circuits compared with DNA-based ones, and second is the advance over Wroblewska et al. (*Nat Biotechnol.*, 2015).

Regarding the advantages of RNA-based logic circuits, almost all studies on gene logic circuits to which the reviewer referred utilize DNA. DNA-based circuits have great potential for cell lineage tracing systems and designer cells (e.g., CAR-T cells). However, RNA-based circuits are much safer and therefore have more potential in therapeutic applications, especially cell therapies in the field of regenerative medicine, such as the elimination of unwanted cells and the purification of desired cells from a heterogenous population differentiated from human pluripotent stem cells. The greater safety is because synthetic mRNAs have less risk than DNAs of genomic integration into the target cells. We believe that making logic circuits with an RNA-only delivery method (i.e., no DNA delivery) is important for these applications.

As for the advance over Wroblewska et al., we would like to emphasize that the construction of any logic circuits shown in this study (AND, OR, NAND, NOR, and XOR gates) has not been achieved previously. In addition, we could not construct logic circuits by simply combining existing repressor devices used in the previous work. As shown in Figure 2, we improved the response of the circuits by redesigning the miRNA target sites. Additionally, we designed and engineered a new MS2CP-responsive mRNA device to increase the ON/OFF ratio in outputs, by stabilizing the RNA secondary structure and surrounding sequence of the binding motif (newly added data in Supplementary Figure 5). By doing so, it became possible for the first time to construct multi-input logic circuits with RNA-only delivery, owing to the improvements of the miRNA- and protein-responsive devices. The basic structure of the devices and network topology of logic gates in this study are well standardized and characterized. Development of reliable standard logic gates is a key step toward construction of a scalable and more complex circuit. Thus, we believe that the construction of RNA-only logic circuits

in this study provides an important milestone, which leads to future diagnostic and therapeutic applications and shows the advancement over the previous work.

Thus, we revised the Introduction, Results and Discussion to help readers understand the significance of our work. Especially, we added new paragraph in Discussion about the advance over the previous works as mentioned above. We also added new experimental data to show the improvement of our MS2CP-responsive device (Supplementary Figure 5). We hope these revisions meet the reviewer's concern.

Reviewers' Comments:

Reviewer #1:

Remarks to the Author:

All my comments were addressed and the current manuscript is recommended for publication as is.